# Undesired Effect of Vancomycin Prolonged Treatment: Enhanced Biofilm Production of the Nosocomial Pathogen *Candida auris*

**DOI:** 10.3390/antibiotics11121771

**Published:** 2022-12-08

**Authors:** Angela Maione, Alessandra La Pietra, Maria Michela Salvatore, Marco Guida, Emilia Galdiero, Elisabetta de Alteriis

**Affiliations:** 1Department of Biology, University of Naples Federico II, 80126 Naples, Italy; 2Department of Chemical Sciences, University of Naples Federico II, 80126 Naples, Italy; 3Institute for Sustainable Plant Protection, National Research Council, 80055 Portici, Italy; 4BAT Center-Interuniversity Center for Studies on Bioinspired Agro-Environmental Technology, University of Naples Federico II, 80055 Portici, Italy

**Keywords:** *Candida auris*, fungal infections, biofilm, vancomycin, prolonged treatment

## Abstract

Fungal infections are often consequent to prolonged antibiotic treatments. Vancomycin (Van) is the first-choice antibiotic in the treatment of *Staphylococcus aureus* infections associated with colonization of catheter surfaces. We demonstrate the direct effect of Van in promoting the formation of the biofilm of the emergent yeast pathogen *Candida auris,* developed in the conventional polystyrene microwell plate model, as well as on silicone surfaces (22 and 28% increase in total biomass, respectively) and on an *S. aures* biofilm, residual after vancomycin treatment, where *C. auris* achieved 99% of the mixed biofilm population. The effect of Van was assessed also in vivo, in the *Galleria mellonella* infection model, which showed higher mortality when infected with the yeast pathogen in the presence of the antibiotic. This evidence enhances awareness of the potential risk associated with prolonged antibiotic use in promoting fungal infections.

## 1. Introduction

In the past decades, invasive fungal infections (IFIs) have emerged as one of the most serious health problems causing high morbidity and mortality in immunocompromised and/or intensive care unit patients. These infections are often subsequent to a prolonged antibiotic treatment, which elicits a shift from commensalism to pathogenicity, as in the case of *Candida* spp. and some filamentous fungi [1]. Most infections are associated with biofilm formation, occurring on both biotic or abiotic surfaces, the latter represented by implanted catheters and other medical devices. In particular, utilization of the central venous catheter is related to a higher risk of developing catheter-related bloodstream infections (CRBI) in intensive care unit patients [2].

Among the different microorganisms able to colonize indwelling catheters forming highly structured biofilms, *Staphylococcus aureus* represents one of the most commonly found, with over 90% of the strains resistant to methicillin (methicillin resistant *S. aureus,* MRSA), penicillin, aminoglycosides, macrolides, lincosamides, and other beta-lactams [3,4]. Vancomycin (Van) is considered a first-choice antibiotic for the treatment of methicillin-resistant *S. aureus* (MRSA) infections [5,6], with development of resistance relatively rare. The high molecular weight (1449.2 g mol^−1^) and hydrophilicity hinder its penetration into biofilms [7,8,9], so that the minimal biofilm inhibition concentration (MBIC) and the minimal biofilm eradication concentration (MBEC) values of Van against *S. aureus* were at least 8 and 16 times higher than the minimal inhibition concentration (MIC) [8,10].

It was reported that some antibiotics have a direct effect on growth and biofilm formation of fungal species such as *Candida* spp. and *Trichosporon* spp. In particular, rifampicin, an antibiotic shown to be capable of inducing *MDR1* expression in *C. albicans,* was proved to promote the formation of the fungal biofilm [11].

More recently, some authors [12] have shown that amoxicillin and cefepime stimulate biofilm formation of *C. albicans*. Further, Van was shown to enhance growth and biofilm formation of the filamentous fungus *Trichosporon* spp., also increasing its virulence in the model organism *Caenorhabditis elegans* [13]. However, the possible action mechanism and molecular targets of Van have not been yet investigated.

The direct effect of antibiotics on the emergent pathogen yeast *Candida auris* has not been reported so far. *C. auris* is a human pathogen of concern according to all governmental institutions (Centers for Disease Control and Prevention (CDC); European Centre for Disease Prevention and Control (ECDC); World Health Organization (WHO); Pan American Health Organization (PAHO); National Institute for Communicable Diseases (NICD)), which have provided guidelines for its diagnosis, clinical treatment, and control since 2016 [14]. Indeed, *C. auris* is associated with severe invasive infections with high mortality rates ranging from 35 to 72% [15,16,17,18]. Among the several virulence traits characterizing *C. auris*, the ability to form biofilms highly tolerant to antifungals has been reported [19,20,21]. Further, in our laboratory the presence of persisters in *C. auris* biofilms has been recently detected [22].

Here, aiming to evaluate the possible risk of a prolonged antibiotic therapy on the development of *C. auris* infections, we investigated the effect of Van on the formation of *C. auris* biofilm, developing in vitro on polystyrene and silicone surfaces, and notably, on a remaining *S. aureus* biofilm, resulting after Van eradication. In addition, the effect of Van on *C. auris* in vivo was evaluated by monitoring the survival of larvae of *Galleria mellonella* (greater wax moth) infected with the yeast pathogen in the presence of the antibiotic.

## 2. Results

### 2.1. Van Increases C. auris Biofilm on Polystyrene and Silicon Surfaces

The effect of three different antibiotics, namely amoxicillin, vancomycin, and rifamycin, administered at the plasma peak (PP) and half plasm peak (PP/2) concentrations, was investigated on the formation of *C. auris* biofilm in the polystyrene microwell plate (Figure 1). For Van, PP and PP/2 corresponded to 30 and 15 g mL^−1^: A significant increase in both the total and viable biofilm biomass (22 and 14%), as determined by the crystal violet method and viable cell count, respectively, was observed when the biofilm was allowed to form in the presence of Van at PP concentration.

The Van effect was confirmed in the case of a *C. auris* biofilm developed on a silicon surface when viable yeast biomass was determined, after scraping the adherent biofilm from the support, resulting 28% higher if the biofilm development was allowed to occur in the presence of Van (Figure 2).

The effect of Van on *C. auris* biofilm growth was also shown by the increased tolerance of the mature biofilm developed in the presence of Van at PP concentration towards the antifungal caspofungin (CAS) (Figure 3). The tested concentrations of caspofungin (from 2.5 to 50 μg mL^−1^) used to eradicate the yeast biofilm were less effective if the latter was allowed to form in the presence of Van, providing indirect evidence of a more robust biofilm developed in the presence of the antibiotic.

To investigate the effect of Van at PP concentration during the different phases of *C. auris* biofilm formation, the antibiotic was added to the culture medium at the beginning of biofilm formation (adhesion), after 24 h (development) or after 48 h (maturation). Incubation with Van was for 6 h in the first case or 24 h in the other two, always prolonging biofilm formation for 72 h. Results are reported in Figure 4, which shows a significant increase in biofilm biomass (total and viable) when Van was added after 48 h, that is, during biofilm maturation.

### 2.2. Vancomycin Enhances C. auris Virulence In Vivo

Larvae of *G. mellonella* represent a valid alternative model for testing both pathogen virulence and antimicrobial compounds. We checked the effect of Van on *C. auris* virulence by infecting larvae with the pathogen in the presence of the antibiotic. Larvae survival was estimated within 72 h, and the results are reported in Figure 5. 

Van at PP concentration significantly increased larvae mortality after 24 h, since survival of larvae infected with *C. auris* with Van was only 30% compared to 75% of the group infected with *C. auris* alone. At longer times (48 and 72 h), larvae survival was very low (10% and 5%, respectively) for both larvae infected with *C. auris* alone or *C. auris* with Van.

### 2.3. Van Favors C. auris Colonization on a Residual S. aureus Biofilm

The *S. aureus* result showed susceptibility to Van, with a MIC value of 1 μg mL^−1^ (Figure 6, left panel). Van was able to eradicate the biofilm of *S. aureus* developed in the microwell plate (Figure 6, right panel). Indeed, eradication was about 95 and 98% at concentrations of 10 and 30 µg mL^−1^ corresponding to 10 × MIC and 30 × MIC, respectively. 

Afterwards, *C. auris* was allowed to colonize the residual *S. aureus* biofilm, remaining after eradication with Van at PP concentration (30 × MIC), giving rise to a mixed bacterial–fungal biofilm. If yeast colonization occurred in the presence of Van, the resulting biofilm contained a higher percentage of *C. auris* (99%) and a negligible percentage of *S. aureus* compared to the biofilm formed in the absence of Van (Figure 7). This indicated that *C. auris* colonization was favored by the presence of the antibiotic. From the result, it was not possible to discriminate between the direct effect of Van on *C. auris* biofilm formation, as shown in the case of the mono-specific biofilm (Figure 1, Figure 2, Figure 3 and Figure 4), and the indirect effect due to the inhibitory effect of Van on *S. aureus* growth, which unavoidably favored the predominance of *C. auris* in the resulting mixed biofilm.

### 2.4. Clues to the Possible Action Mechanism

To start the investigation on the possible action mechanism of Van in promoting *C. auris* biofilm growth, we determined by qRT-PCR the expression of some genes in the cells collected from a mature biofilm developed in the presence of Van and compared it with the expression of the same genes in the cells collected from a biofilm developed in the absence of Van. The selected genes were *ERG11*, one of the genes involved in ergosterol biosynthesis, *ALS5* coding for an adhesin, *FKS1* coding for the β-1,3-glucan synthase, a key enzyme to synthesize an essential component of the fungal cell wall, and *HOG1*, a stress-related gene.

Figure 8 represents the relative changes, compared to the biofilm cells grown in the absence of Van, in the expression levels of the genes, indicated as fold change values and normalized to the housekeeping gene *ACT1*. The *FKS1* and *HOG1* genes results were slightly upregulated, but notably, the *ERG11* gene was twice upregulated, this suggesting a possible involvement of Van in ergosterol biosynthesis.

Starting from this observation, we hypothesized that Van could enhance ergosterol biosynthesis favoring biomass production when growth occurred in conditions of biofilm development. Hence, in this study the effect of Van on the sterol profile in sessile *C. auris* was investigated. The percentage of ergosterol was 0.10 ± 0.02 in the control sample, and no significant difference in its content was observed after 48 h exposure to Van (0.08% ± 0.01). Hence, the Van capacity to enhance the biofilm formation of *C. auris* seems to be not linked to direct alterations of membrane sterol profile.

## 3. Discussion

Microbial biofilms routinely develop on surfaces of implanted medical devices. A close connection between fungal infections and the increased use of medical devices has been documented, with a particular alert concerning CRBI following the use of intravascular catheters [23,24,25]. 

In antibiotic lock therapy (ALT), Van is one of the most commonly used antibiotics to sterilize catheters infected by *S. aureus* and other Gram-positive bacterial species. Undoubtedly, ALT represents a form of prolonged antibiotic treatment that is not free from risks, among which is the possible colonization of the catheter by competitor fungal species.

Indeed, some authors have reported the direct effect of some antibiotics on growth and biofilm formation for some fungal species such as *Candida* spp. and *Thrichosporon* spp. [12,13].

In this work, we analyzed the possible stimulating effect of Van, used at PP concentration, on the formation of biofilms of the emergent yeast pathogen, *C. auris*, characterized by its ability to colonize surfaces in the form of biofilms highly tolerant to antifungals [26].

We clearly showed that the formation of *C. auris* biofilm is promoted by the presence of Van in the culture medium. The effect was displayed in vitro, on the biofilm growing on both polystyrene and silicone surfaces. Silicon is a material commonly used in indwelling devices, since it is available as flat medical grade sheeting and is reported to promote *Candida* biofilm formation [27]. Further, we investigated if such biofilm was more tolerant to caspofungin, an antifungal commonly used against *C. auris* [28]. We found less susceptibility to caspofungin for the Van-grown biofilm, this result representing indirect evidence of a more robust aggregate, since increased tolerance to an antifungal is largely associated to biofilm thickness.

*Galleria mellonella* larvae are a model organism to study in vivo microbial infections [29]. We found that the mortality of larvae after 24 h from an injection of *C. auris* was greatly increased by the administration of the yeast pathogen with Van, suggesting that the antibiotic may promote fungal growth and/or virulence also in vivo. This result is in accordance to the observed increase in virulence of *Thrichosporon* spp. in the nematode *C. elegans* when infected in the presence of Van [13]. 

Further, we showed that *C. auris* was able to colonize a residual *S. aureus* biofilm and that yeast colonization was favored by Van. The experiment performed in the conventional microwell plate mimicked the potential undesired effect of a Van treatment aimed at sterilizing an infected catheter but able to promote a fungal super-infection by *C. auris*.

The mechanisms of action of the different antibiotics that are reported to stimulate fungal growth have not been deeply investigated so far. Only in the case of rifampicin, it was found that this compound upregulated *MDR1* gene expression in *C. albicans* [30], and other key virulence determinants such as fibronectin binding, cell hydrophobicity, and germ tube formation, potentially supporting biofilm formation [11]. 

Our results clearly show that Van effect was evident in the developing biofilm after 48 h and that the *ERG11* gene in *C. auris* biofilm cells grown in the presence of Van was significantly upregulated. *ERG11* is involved in ergosterol biosynthesis, an essential component of fungal membranes. Ergosterol is known to regulate membrane fluidity, permeability, and transport [31]. It may also affect the absorption and utilization of nutrients, so that ergosterol has been regarded as a ‘fungal hormone’, able to stimulate the growth and proliferation of fungi [32]. 

Therefore, we hypothesized that Van may act as a survival factor for *C. auris*, in the hypoxic conditions of biofilm maturation, promoting ergosterol biosynthesis. However, despite a significant *ERG11* upregulation, no statistically significant difference in the ergosterol content was found in the membrane of biofilm cells grown in the presence of Van. Hence, Van does not enhance biofilm formation affecting sterol content in *C. auris* membranes, so that further investigations are needed to shed light on the mechanism of action.

One limitation of the present study is represented by the use of the reference *C. auris* strain only; the inclusion of clinical strains, particularly isolated from catheter surfaces, could significantly enhance the significance of the results.

In conclusion, taken together, the outcomes of this study show a direct effect of Van in promoting pathogenic *C. auris* biofilms and reinforce awareness of the possible risk associated with a prolonged use of antibiotics in the onset of fungal infections.

## 4. Materials and Methods

### 4.1. Drugs

Amoxicillin (LGC Standards S.r.l., Sesto San Giovanni, MI, Italy), rifampicin (ChemCruz Biochemicals, Huissen, The Netherlands), and vancomycin (PanReac AppliChem, Barcelona, Spain) were used in this study. Amoxicillin and vancomycin were solubilized in a sterile water solution; rifampicin was solubilized in dimethyl sulfoxide (DMSO). Stock solutions of each drug were prepared at 1 mg mL^−1^. In the experiments, drugs were used at concentrations corresponding to plasma peak (PP) and half plasma peak (PP/2) concentrations.

### 4.2. Strains, Media, and Cultural Conditions

In this study, *C. auris* DSM 21092 and *S. aureus* ATCC 6538 were used. *C. auris* was grown on Rose Bengal Agar (RBA) (VWR Chemicals, Leuven, Belgium) and cultured in Tryptic Soy Broth (TSB) (Oxoid, Basingstoke, UK) with 1% glucose for 18 h at 37 °C and 200 rpm to prepare biomass inoculum for both the biofilm formation. *S. aureus* was grown on Tryptic Soy Agar (TSA) (Oxoid, Basingstoke, UK) and cultured in TSB for 18 h at 37 °C and 200 rpm for inocula preparation. Cells were harvested by centrifugation at 5000 rpm for 5 min at 4 °C and washed twice in NaCl 0.9%. For biofilm formation, RPMI 1640 medium (Thermo Fisher Scientific, Waltham, MA, USA) 50% *v*/*v* was used. In the case of *S. aureus* biofilm, 10% *v*/*v* Fetal Bovine Serum (FBS) (Sigma-Aldrich Co., St. Louis, MO, USA) was added to RPMI.

Petri plates of RBA (VWR Chemicals, Leuven, Belgium) and Baird-Parker Agar (BDA) (OXOID, Basingstoke, UK) were used for total cell counts of *C. auris* and *S. aureus*, respectively.

### 4.3. C. auris Biofilm Formation In Vitro

#### 4.3.1. Effect of Van and Other Antibiotics on *C. auris* Biofilm Formation in Microwell Polystyrene Plate

To evaluate the effect of antibiotics on biofilm formation of *C. auris*, yeast cell suspension (1 × 10^6^ cell mL^−1^) was prepared in RPMI 1640 medium 50% *v*/*v* and 100 µL added in each well of a 96-well polystyrene microplate.

Drugs were added individually to the cell suspension at two different concentrations, 4–2 µg mL^−1^ for amoxicillin (, 30–15 µg mL^−1^ for vancomycin, 40–20 µg mL^−1^ for rifampicin, which corresponded to plasma concentration PP and PP/2 for each drug, respectively. The plate was incubated at 37 °C for 72 h, renewing the culture medium with the antibiotic every 24 h. Control wells where the biofilm was allowed to develop in the same conditions but without the antibiotic were prepared. 

#### 4.3.2. Effect of Van on Different Phases of *C. auris* Biofilm Development

To evaluate the effect of Van on different phases of the fungal biofilm development, Van at PP concentration was added to 6, 24, and 48 h sessile cells of the biofilm previously grown in microtiter plates as previously described, and biofilms were incubated for up to 72 h at 37 °C. 

#### 4.3.3. Susceptibility of *C. auris* Biofilm to Caspofungin

The susceptibility of a *C. auris* biofilm grown in the presence of Van PP to the antifungal caspofungin (CAS) (Sigma-Aldrich Co, St. Louis, MO, USA) was evaluated. Aiming at this, the *C. auris* biofilms were grown for 72 h, as previously described with or without Van at PP concentration, washed with NaCl 0.9%, and treated with CAS (ranging from 2.5 to 50 µg mL^−1^). After 24 h further incubation, total biofilm biomass was quantified. 

### 4.4. Determination of Biofilm Total Biomass

The crystal violet (CV) staining method [33] was used to quantify total biofilm mass. To determine the number of viable cells in the biofilm, a colony forming unit (CFU) assay was performed. Briefly, adhered cells were completely scraped and serially diluted in Phosphate Buffered Saline (PBS). Cell diluted suspensions were spread on an RBA plate supplemented with chloramphenicol and incubated at 37 °C for 48 h. The number of *C. auris* viable cells was reported as Log CFU per well.

### 4.5. C. auris Biofilm Formation on Silicone Surface

Silicon elastomer (SE) sheets (0.5 mm thick) were purchased from Novotema S.p.a. (Villongo, BE, Italy). SE platelets of 1.2 × 12 cm were obtained by cutting with a precision cutter. The material was cleaned by washing in hand soap and water, rinsed with distilled water, and autoclaved. To allow *C. auris* biofilm formation, the procedure by Kuhn et al. [34] was followed with some modifications. The SE platelets were placed in 12-well tissue culture plates and incubated in FBS for 24 h at 37 °C on a rocker table (pre-treatment phase). The platelets were then moved to new plates and washed with PBS to remove residual FBS. To ensure uniform biofilm formation, we immersed them in a *C. auris* cell suspension (1 × 10^7^ cells mL^−1^), and the plate was incubated at 37 °C for 90 min under shaking (adhesion phase). After this period, the platelets were moved into new 12-well plates to ensure the removal of non-adherent cells, each well containing 2 mL RPMI medium 50% *v*/*v* with or without Van at PP concentration and incubated under gentle shaking at 37 °C for 72 h, renewing medium every 24 h (biofilm formation phase). For controls, platelets were processed in identical fashion, except that no *Candida* cells were added. 

To determine the number of adherent cells of the biofilm, the SE platelets were washed in PBS to remove non-adherent cells and carefully scraped, then the CFU assay was performed. The number of *C. auris* viable cells was reported as Log CFU per cm^−2^ of silicone surface.

### 4.6. S. aureus Biofilm Formation and Eradication with Van

#### 4.6.1. Susceptibility of *S. aureus* towards Van

To determine the MIC of Van against *S. aureus*, the CLSI guidelines [35] were considered with some modifications. Briefly, 100 μL of *S. aureus* suspension (1 × 10^6^ cell mL^−1^) in RPMI 1640 medium 50% *v*/*v* with 10% FBS was added to the 96-well plate with different concentrations of Van (ranging from 0.5 and 30 µg mL^−1^). The plate was incubated at 37 °C for 24 h. Bacterial growth was determined by measuring absorbance at 590 nm using a microplate reader (Synergy™ H4; BioTek Instruments, Inc., Winooski, VT, USA). 

#### 4.6.2. *S. aureus* Biofilm Formation 

In order to allow *S. aureus* biofilm development, 100 μL of cell suspension (1 × 10^6^ cell mL^−1^) in RPMI 1640 medium 50% *v*/*v* with 10% FBS, was added to the 96-well plate and incubated at 37 °C for 48 h, renewing the culture media after the first 24 h. The biofilm was washed with NaCl 0.9% to remove non-adherent cells. Total biofilm biomass was quantified with the crystal violet staining methodology [33].

#### 4.6.3. Eradication with Van of *S. aureus* Biofilm

In order to analyze the Van eradication effect on *S. aureus* mature biofilm, Van (10, 30, and 50 µg mL^−1^) was added to a mature 48 h *S. aureus* biofilm, prepared as described above. The biofilm was washed with NaCl 0.9%, disrupted by scraping and vortexed for 30 s. Serial dilutions were plated on BDA plates and after incubation at 37 °C for 24 h, the number of bacterial colonies was determined. Residual *S. aureus* biofilm was calculated as a percentage of the control biofilm. 

### 4.7. Effect of Van on C. auris Growth on a Residual S. aureus Biofilm

To investigate the effect of Van on *C. auris* biofilm formation on a residual *S. aureus* biofilm, a 48 h *S. aureus* biofilm was treated with Van at PP concentration. Then, the microwell plate was abundantly washed and a suspension of *C. auris* (100 μL of 1 × 10^6^ cell mL^−1^) in RPMI 50% *v*/*v* was added, alone and in combination with Van PP on the residual *S. aureus* biofilm. The plate was incubated for another 72 h at 37 °C, renewing the RPMI medium 50% *v*/*v* with and without Van PP every 24 h. After this period, the composition of biofilm was determined after scraping the wells and plating the adherent cells on RBA and BDA, to discriminate bacterial from fungal colonies, respectively. 

### 4.8. In Vivo Effect of Van on G. mellonella Larvae

To analyze the effect of Van in vivo, *G. mellonella* larvae were used. For each group, 30 randomly chosen larvae with bodyweights of approximately 300 mg were used for each test group. One group was used to test the toxicity of Van at PP concentration, 10 μL of which were injected in the last left pro-leg of the larva. Another group was infected with 10 μL of cell suspension (1 × 10^6^ cell mL^−1^) of *C. auris* alone and another with 10 μL of cell suspension in combination with Van at PP concentration. Two groups were used as control: one untreated (intact larvae), and one treated with 10 μL of PBS. The larvae were incubated at 37 °C in the dark and survival was monitored for 3 days. 

### 4.9. qRT-PCR

The gene expression of *ALS5*, *ERG11*, *FSK1,* and *HOG1* was analyzed using qRT-PCR. Biofilm was grown with or without Van (control) at PP concentration, as previously described, and then was washed with NaCl 0.9%, scraped and collected in centrifuge tubes. After centrifugation at 4 °C for 5 min at 4500 rpm, the supernatant was discarded and the pellet was used to extract total RNA. Total RNA was extracted and purified using Direct-zolTM RNA Miniprep Plus Kit (ZYMO RESEARCH, Irvine, CA, USA) and quantified using Nanodrop spectrophotometer 2000 (Thermo Fisher Scientific Inc., Waltham, MA, USA). An iScriptTM cDNA Synthesis Kit (BioRad, Milan, Italy) was used to retrotranscribe 1000 ng of RNA for each sample following manufacturer’s instructions. A total of 1 µL cDNA was used as a template in the reaction (final volume 10 µL) with 0.3 mM of each primer (Table 1) and 1× SensiFAST SYBR Green master mix (Meridiana Bioline, Italy) for qRT-PCR, which was performed in AriaMx Real-Time PCR instrument (Agilent Technologies, Inc., Santa Clara, CA, USA) using the following thermal profile: 95 °C for 10 min, one cycle for cDNA denaturation; 95 °C for 15 s and 60 °C for 1 min, 40 cycles for amplification; 95 °C for 15 s; one cycle for final elongation; and one cycle for melting curve analysis (from 60 to 95 °C). To measure the fluorescence, the Agilent Aria 1.7 software (Agilent Technologies, Inc., Santa Clara, CA, USA) was used. The REST software (Relative Expression Software Tool, Weihenstephan, Germany, version 1.9.12), based on the Pfaffl method [36] was used to analyze each gene and its expression was normalized to the standard *ACT1* gene. Fold changes respect to control greater than ±1.0 were considered significant.

### 4.10. Ergosterol Extraction and Quantitation

Ergosterol extraction and quantitation were carried out according to Rahimi-Verki et al. [37] with some modifications. Following the cultivation for 48 h in the presence or absence of Van (PP concentration), biofilm cells of *C. auris* were rapidly and abundantly washed with distilled water. The net weight of the cell pellet (≈20 mg d.w.) was determined. 3 mL of 25% alcoholic potassium hydroxide solution was added to each pellet and vortex mixed for 1 min. The cell suspensions were transferred to sterile borosilicate glass screw-cap tubes and incubated for 3 h in a water bath of 80 °C. Following incubation, the tubes were allowed to cool. Sterol was extracted by addition of a mixture of 1 mL of sterile distilled water and 3 mL of hexane followed by vigorous vortex mixing for 3 min. The hexane layer was transferred to a clean borosilicate glass screw-cap tube and stored at −20 °C. A 5-fold dilute solution of sterol extract was prepared in absolute ethyl alcohol and scanned spectrophotometrically from 200 to 300 nm by V-750 Spectrophotometer by Jasco Europe s.r.l. (Lecco, Italy). Finally, the amount of ergosterol in each control and treated tube was obtained using Equation (1), where F is the factor for dilution in ethanol and 290 and 518 are the E (percent per centimeter) values determined for crystalline ergosterol and 24(28)dehydroergosterol, respectively [38].
(1)$$\%\;\mathrm{Ergosterol}=\left[\frac{\left(A_{281.5}/290\right)\times F}{\mathrm{sample}\;\mathrm{weight}}\right]-\left[\frac{\left(A_{230}/518\right)\times F}{\mathrm{sample}\;\mathrm{weight}}\right]$$

### 4.11. Statistical Analysis

The results and data were obtained by three independent experiments and represented as mean values ± standard deviation (SD). For data analysis, GraphPad Prism Software (version 8.02 for Windows, GraphPad Software, La Jolla, CA, USA, www.graphpad.com, accessed on 16 September 2022) was used. Biofilm formation and eradication data were analyzed using one-way and two-way analysis of variance (ANOVA) followed by Sidak’s test. The Student’s *t*-test was used for the data of biofilm formation on silicone surfaces and *C. auris* growth on the residual *S. aureus* biofilm. Survival curves were plotted using the Kaplan–Meier method, and the differences between groups were analyzed using Tukey’s multiple comparation test. Asterisks (* = *p* < 0.05, ** = *p* < 0.01, *** = *p* < 0.001, **** = *p* < 0.0001) indicate significant differences.

## Figures and Tables

**Figure 1 antibiotics-11-01771-f001:**
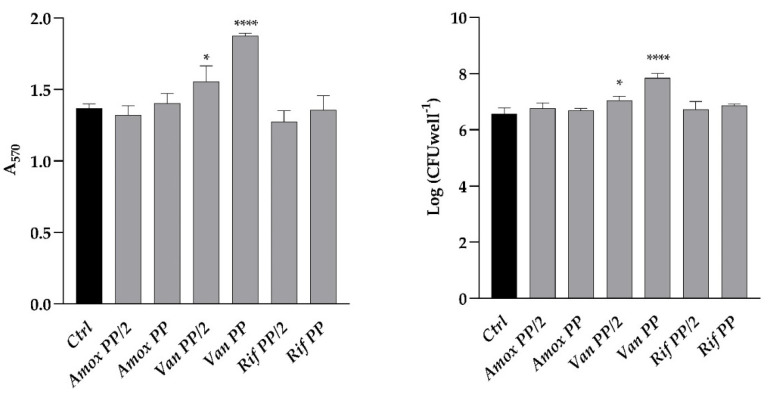
Comparison between biofilm formation of *C. auris* with different drugs (Amox, amoxicillin; Van, Vancomycin; Rif, rifampicin) at PP and PP/2 concentrations as compared to control (Ctrl). (**Left panel**) Total biomass evaluated by crystal violet staining (A_570_); (**Right panel**) Log of colony-forming unit (CFU) per well. Significant differences were determined by One way ANOVA (Sidak’s test): * *p* < 0.05; **** *p* < 0.0001.

**Figure 2 antibiotics-11-01771-f002:**
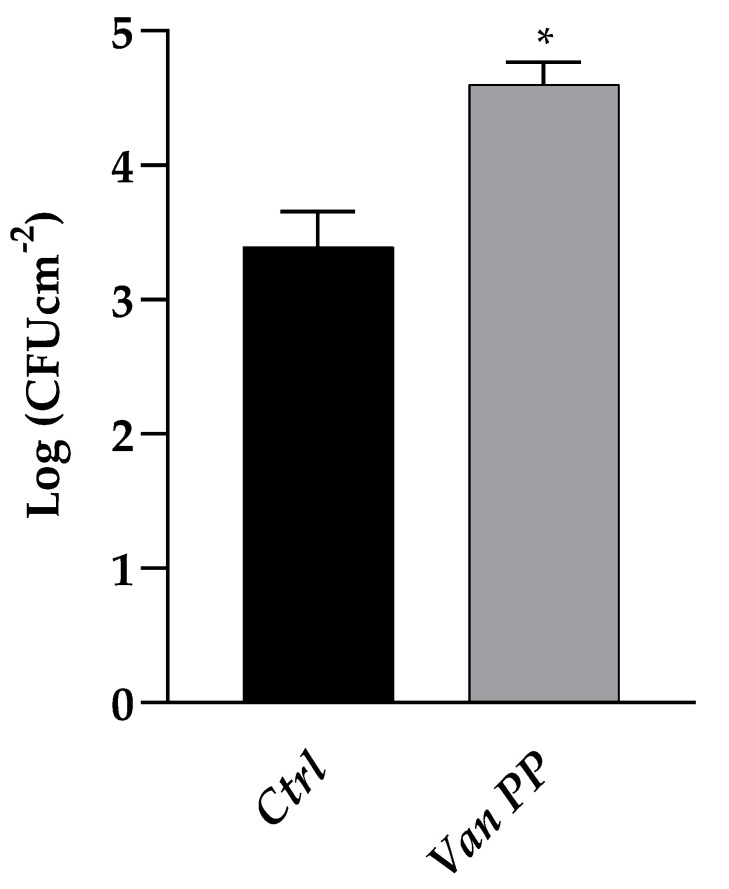
Comparison between *C. auris* biofilm formation on silicon platelets, with Van at PP concentration as compared to control (Ctrl). Statistical significance: * *p* < 0.05 (Student’s *t*-test).

**Figure 3 antibiotics-11-01771-f003:**
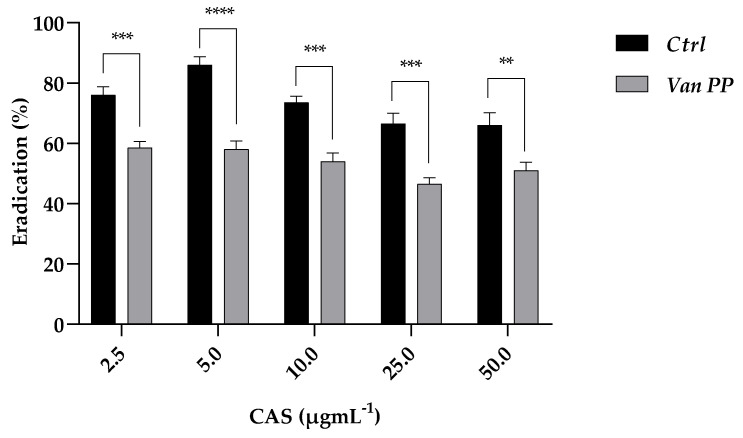
Eradication of mature *C. auris* biofilm formed with and w/o Van PP (Ctrl) with caspofungin (CAS) at 2.5–50 μg mL^−1^ concentrations. Significant differences were determined by Sidak’s test: ** *p* < 0.01; *** *p* < 0.001; **** *p* < 0.0001.

**Figure 4 antibiotics-11-01771-f004:**
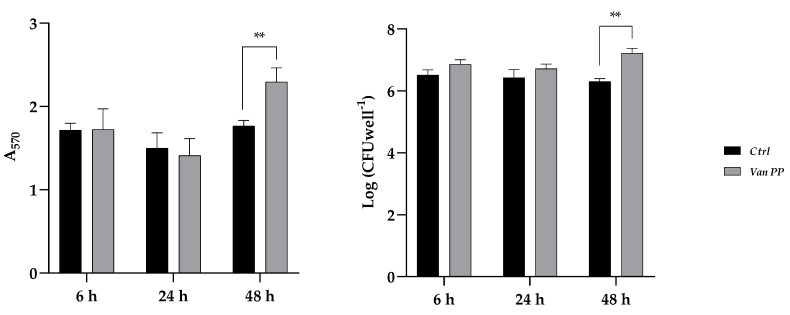
*C. auris* biofilm in the presence of Van at PP concentration added after 6, 24, and 48 h during biofilm formation. (**Left panel**) Total biomass evaluated by crystal violet staining method (A_570_); (**Right panel**) Log of colony-forming unit (CFU) per well. Statistical significance: ** *p* < 0.01 (Sidak’s test).

**Figure 5 antibiotics-11-01771-f005:**
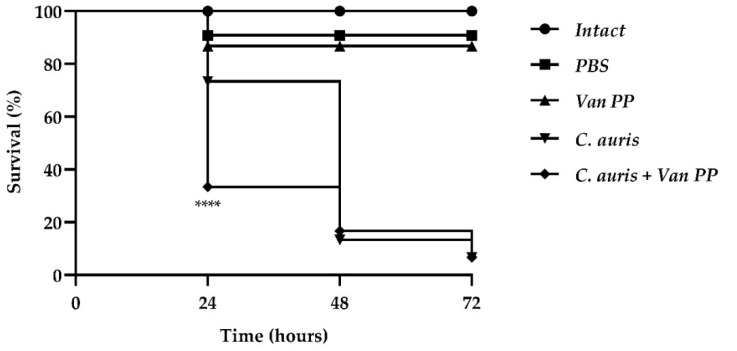
Survival of *G. mellonella* larvae represented by Kaplan–Meier plot. Significant differences were determined by Tukey’s test (**** *p* < 0.0001) between groups infected with *C. auris* alone and *C. auris* with Van PP.

**Figure 6 antibiotics-11-01771-f006:**
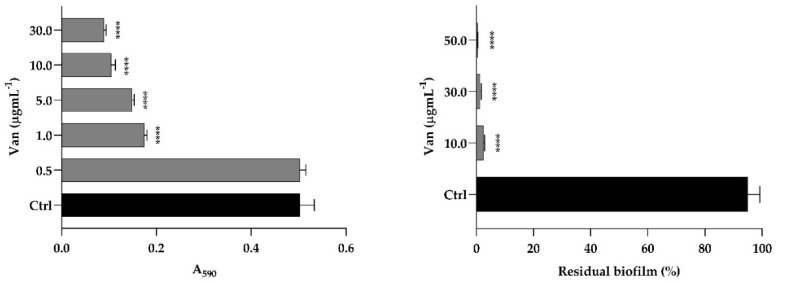
(**Left**) Susceptibility of *S. aureus* planktonic cells to Van (0.5; 1.0; 5.0; 10.0; 30.0 µg mL^−1^); (**Right**) Eradication of *S. aureus* biofilm with Van (10.0; 30.0; 50.0 µg mL^−1^) expressed as residual biofilm and compared to control (Ctrl). Significant differences were determined by Sidak’s test: **** *p* < 0.0001.

**Figure 7 antibiotics-11-01771-f007:**
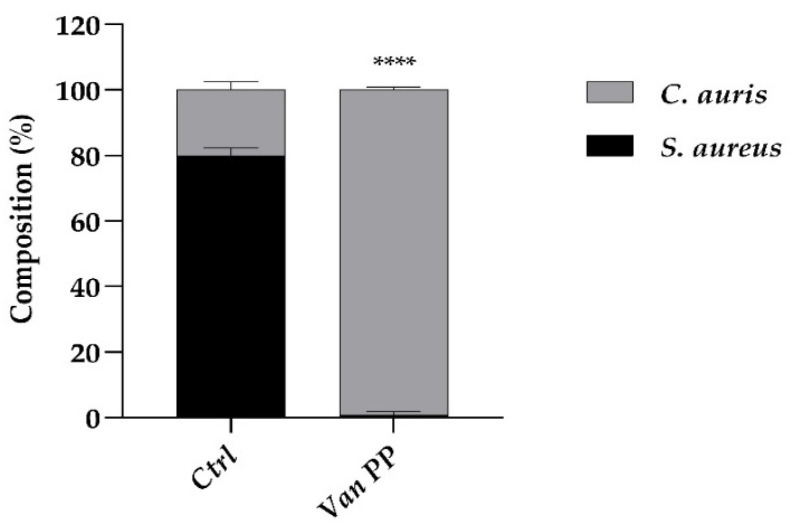
Comparison between *C. auris* growth with and w/o Van at PP concentration (Ctrl) on an *S. aureus* biofilm, previously eradicated with Van. Cell composition in the mixed biofilm resulting after eradication was represented as percentage (%) of *S. aureus* and *C. auris* cells in the biofilm. Statistical significance: **** *p* < 0.0001 (Student’s *t*-test).

**Figure 8 antibiotics-11-01771-f008:**
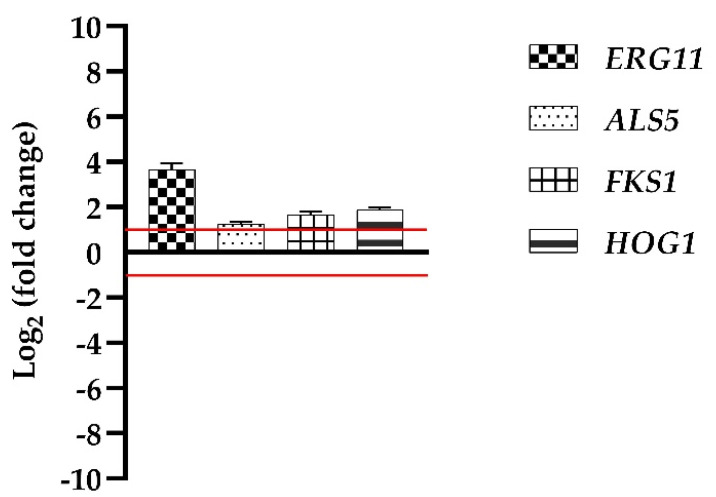
Real-time qPCR. Histograms represent the fold differences in the expression levels of four genes, *ERG11*, *ALS5*, *FKS1*, and *HPG1*, in biofilm cells developed with Van compared to biofilm cells w/o Van. Red lines show fold change thresholds of −1 and +1, respectively.

**Table 1 antibiotics-11-01771-t001:** Gene-specific primers used for real-time RT-PCR.

Gene Name	Acronym	GenBank Accession	Primer Name	Sequence (5′->3′)
** *Actin* **	*actin*	NW_021640162	*C. auris_actin_F*	GAAGGAGATCACTGCTTTAGCC
			*C. auris_actin_R*	GAGCCACCAATCCACACAG
** *Hyphal-specific genes* **	*ALS5*	XM_712981	*C. auris_ALS5_F*	CCTTCTGGATCGGACACAGT
			*C. auris_ALS5_R*	AGTTGTGGTGGAGGAACCAG
** *Ergosterol Biosynthesis* **	*ERG11*	KY410388.1	*C. auris_ERG11_F*	GTGCCCATCGTCTACAACCT
			*C. auris_ERG11_R*	TCTCCCACTCGATTTCTGCT
** *1,3-beta-glucan synthase* **	*FSK1*	NW_021640162.1	*C. auris_FSK1_F*	GCAAACTTTCATGTTGGTGTTA
			*C. auris_FSK1_R*	TGTGAACAAGGAGTTTGAGTAA
** *High-osmolarity glycerol1* **	*HOG1*	NW_021640166	*C. auris_HOG1_F*	GACTTGTGGTCTGTGGGTTG
			*C. auris_HOG1_R*	ACATCAGCAGGAGGTGAGC

## Data Availability

Not applicable.

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
