# Peer review of "Undesired Effect of Vancomycin Prolonged Treatment: Enhanced Biofilm Production of the Nosocomial Pathogen Candida auris"

_antibiotics, 2022, doi:10.3390/antibiotics11121771_

Round 1

Reviewer 1 Report

-

Author Response

English language and style have been checked

Reviewer 2 Report

Overall appreciation

This is a well-conducted study on a very relevant subject nowadays. Congratulations on the well-designed work. Despite this study will benefit from the inclusion of clinical strains, particularly isolated from catheter surfaces, the study still presents merit. However, this limitation must be disclosed in the conclusion, as the results may be significantly changed in the presence of clinical isolates. Still, the study brings new insights into the modulation of biofilm formation in the presence of vancomycin.

Minor considerations are detailed below.

Abstract

After defining the acronym van, please use it in the abstract text.

Please include specific result values in the abstract regarding biofilm promotion on the used models.

Introduction

Line 38: Remove the dot after Staphylococcus

Line 44: Please define MBIC and MBEC

Uniformize the use of C. ou Candida across the text

Results

2.1. Van Increases C. auris Biofilm on Polystyrene and Silicon Surfaces – italicize the word C. auris

Check the text for errors in the italics and abbreviations throughout the manuscript

Methods

Strains, Media, and Cultural Conditions

In what solvent was RPMI diluted to achieve a 50% v/v concentration? And why use this dilution of the culture medium?

4.3 . in this section, 3 variations of the protocol are described with 3 different objectives. Please divide them into three subsections, eg. 4.3.1, 4.3.2, etc, to clarify this section.

Line 299- correct the typo in this line CFU Nether-299 lands)

4.5 – the visual MIC is the recommended output in CLSI guidelines. Did you determine the visual MIC? If so please include the value in the results section. Please consider the comment done for section 4.3, and also for this section.

Results

Line 120 – “Van significantly increased larvae mortality after 24 h, since survival was only 30% 120 compared to 75% of the group infected with C. auris alone.” Please clarify if the text that the first set of results described regards Van + C. auris and not only Van

2.3 – include some text description for the left results at the beginning of the section.

2.4 – if the graphic was interpreted correctly, results only for the treatment group are shown as fold variation. Why not to shown the results for the treated and untreated groups in order to compare them?  Also, the red lines are not in accordance with the legend.

Discussion

There are other methods to quantify ergosterol in your treated and untreated samples, described in the literature? If so the results could have been influenced by the method itself, and others may present advantages. If this is true for your study, please discuss the hypothesis in the discussion.

Author Response

This is a well-conducted study on a very relevant subject nowadays. Congratulations on the well-designed work. Despite this study will benefit from the inclusion of clinical strains, particularly isolated from catheter surfaces, the study still presents merit. However, this limitation must be disclosed in the conclusion, as the results may be significantly changed in the presence of clinical isolates. Still, the study brings new insights into the modulation of biofilm formation in the presence of vancomycin.

We thank the Reviewer for appreciation of the paper. According to the Referee’s suggestion, the limitation  concerning the use of the reference strain of C. auris only has been added to the Discussion (lines…)

Minor considerations are detailed below.

Abstract

After defining the acronym van, please use it in the abstract text.

Done

Please include specific result values in the abstract regarding biofilm promotion on the used models.

Percentage values of total biomass increase have been added to the Abstract and also to Results (lines 81, 92 and 146 in the revised ms).

Introduction

Line 38: Remove the dot after Staphylococcus

Done

Line 44: Please define MBIC and MBEC

Done

Uniformize the use of C. ou Candida across the text

Done

Results

2.1. Van Increases C. auris Biofilm on Polystyrene and Silicon Surfaces – italicize the word C. auris

Done

Check the text for errors in the italics and abbreviations throughout the manuscript

Done

Methods

Strains, Media, and Cultural Conditions

In what solvent was RPMI diluted to achieve a 50% v/v concentration? And why use this dilution of the culture medium?

RPMI was diluted in distilled water. RPMI is a medium containing molecules (sodium bicarbonate and methionine) that can induce the shift from the yeast-like to the hyphal form, so promoting biofilm formation (Yu, Q., Li, J., Zhang, Y. et al.  Sci Rep 6, 26667, 2016). In our lab, we observed that the dilution favours biofilm formation even more, due to the reduced glucose concentration.

4.3 . in this section, 3 variations of the protocol are described with 3 different objectives. Please divide them into three subsections, eg. 4.3.1, 4.3.2, etc, to clarify this section.

Done

Line 299- correct the typo in this line CFU Nether-299 lands)

Done

4.5 – the visual MIC is the recommended output in CLSI guidelines. Did you determine the visual MIC? If so please include the value in the results section. Please consider the comment done for section 4.3, and also for this section.

MIC values were determined according to CSLI guidelines, measuring the absorbance at 590 nm, as described in Materials and Methods.

Following the Reviewer’s suggestion, also section 4.5 has been divided in subparagraphs (4.5.1, 4.5.2, 4.5.3).

Results

Line 120 – “Van significantly increased larvae mortality after 24 h, since survival was only 30% 120 compared to 75% of the group infected with C. auris alone.” Please clarify if the text that the first set of results described regards Van + C. auris and not only Van

Done

2.3 – include some text description for the left results at the beginning of the section.

Done (lines 134-135 in the revised ms)

2.4 – if the graphic was interpreted correctly, results only for the treatment group are shown as fold variation. Why not to shown the results for the treated and untreated groups in order to compare them?  Also, the red lines are not in accordance with the legend.

The REST software used for analysing the expression results directly reports fold variations respect to untreated controls (in our case samples of biofilm grown without Van).

The legend of Figure 8 has been corrected.

Discussion

 There are other methods to quantify ergosterol in your treated and untreated samples, described in the literature? If so the results could have been influenced by the method itself, and others may present advantages. If this is true for your study, please discuss the hypothesis in the discussion.

The determination of ergosterol in this study was conducted using a validated method reported in the literature which means that even if other methods may exist, in abstract, all validated method should give the same results within statistical variations. We respectfully think that further discussion is unnecessary.

English language has been checked.

Reviewer 3 Report

The article „Undesired Effect of Vancomycin Prolonged Treatment: Enhanced Biofilm Production of the Nosocomial  Pathogen Candida auris“ is an interesting and well-written scientific work.

As the research was conducted and written professionally, I have only a few minor remarks:

line 17: to - with

line 18: vancomicyn - vancomycin

line 22: delete a

line 25: full name of C.auris, add Vancomycin, Biofilm,  Prolonged treatment;

line 32: to - with

line 38: delete the extra dot

lines 44-45: introduce MBIC, MBEC and MIC abbreviations

line 46: a direct

line 50: cefepime

line 59: Institute

Results- subsections 2.1; 2.2; 2.3. C.auris-Italic; 2.2 in vivo-italic; 2.3-S aureus-italic

line 183: introduce ALT abbreviation

line 201: an antifungal

line 204: an injection

line 207: in virulence

line 238: a sterile

4.3; 4.4; 4.6. C. auris-italic; 4.5. S.aureus-italic

line 258: the biofilm

line 261: the cell

line 270: for up

line 293: into

line 295: Van; delete the(medium)

line 303: move the introduction of the MIC abbreviation to line 45

lines 330-338: name Larvae groups (e.g. I/II or 1./2.) and connect them with Figure 5 in the Results

line 343: the supernatant

line 349. each primer

line 354: dot after )

line 357: their-its

Author Response

Comments and Suggestions for Authors

The article „Undesired Effect of Vancomycin Prolonged Treatment: Enhanced Biofilm Production of the Nosocomial  Pathogen Candida auris“ is an interesting and well-written scientific work.

As the research was conducted and written professionally, I have only a few minor remarks:

Thank you for appreciation.

All the suggested corrections have been done.

English language and style have been checked.

line 17: to - with

line 18: vancomicyn - vancomycin

line 22: delete a

line 25: full name of C.auris, add Vancomycin, Biofilm,  Prolonged treatment;

line 32: to - with

line 38: delete the extra dot

lines 44-45: introduce MBIC, MBEC and MIC abbreviations

line 46: a direct

line 50: cefepime

line 59: Institute

Results- subsections 2.1; 2.2; 2.3. C.auris-Italic; 2.2 in vivo-italic; 2.3-S aureus-italic

line 183: introduce ALT abbreviation

line 201: an antifungal

line 204: an injection

line 207: in virulence

line 238: a sterile

4.3; 4.4; 4.6. C. auris-italic; 4.5. S.aureus-italic

line 258: the biofilm

line 261: the cell

line 270: for up

line 293: into

line 295: Van; delete the(medium)

line 303: move the introduction of the MIC abbreviation to line 45

lines 330-338: name Larvae groups (e.g. I/II or 1./2.) and connect them with Figure 5 in the Results

line 343: the supernatant

line 349. each primer

line 354: dot after )

 line 357: their-its

Reviewer 4 Report

The original article by Maione and collaborators reports several essential aspects of vancomycin influence on C. auris biofilm facilitation and formation, virulence, and drug resistance, using in vitro and in vivo assays.

Such a report significantly contributes to the community dealing with pathogenic yeasts and drug-resistant microorganisms.

For a definitive publication, minor corrections and some suggestions are recommendable

It’s recommendable to include the common name of  Galleria mellonella

In the “Results”  section:

Include the numeric values of plasma peak (PP) and half plasm peak (PP/2) the first time they appear in the text.

It would be interesting for the authors to include pictures of crystal violet-stained biofilms for visualization with the graphics of biofilm quantitation for better illustration.

With regards to PP/ and PP concentrations:  are these values correct? …at concentrations of 10 µg mL-1. (PP/2) and 30 (PP) µg mL-1. Is PP/2 half of PP? Then, 15 µg mL-1 or PP = 20 µg mL-1

It is recommendable to Define ALT the first time it appears and check all abbreviations

“Materials and methods” section:

In Table 1, “Gene-specific primers used for real-time RT-PCR,” it is recommended to include and list the GenBank access numbers for the selected genes in an extra column.

“Discussion” section:

Have the authors investigated the direct antifungal activity of vancomycin against C.auris? It has been indicated that vancomycin inhibits the growth of planktonic and biofilm Candida spp (β-Lactam antibiotics and vancomycin inhibits the growth of planktonic and biofilm Candida spp.: An additional benefit of antibiotic-lock therapy? International Journal of Antimicrobial Agents,https://doi.org/10.1016/j.ijantimicag.2014.12.012).

Author Response

The original article by Maione and collaborators reports several essential aspects of vancomycin influence on C. auris biofilm facilitation and formation, virulence, and drug resistance, using in vitro and in vivo assays.

Such a report significantly contributes to the community dealing with pathogenic yeasts and drug-resistant microorganisms.

For a definitive publication, minor corrections and some suggestions are recommendable

It’s recommendable to include the common name of  Galleria mellonella

Done (line 74 of the revised ms)

In the “Results”  section:

Include the numeric values of plasma peak (PP) and half plasm peak (PP/2) the first time they appear in the text.

Done (line 80 in the revised ms)

It would be interesting for the authors to include pictures of crystal violet-stained biofilms for visualization with the graphics of biofilm quantitation for better illustration.

We agree with the Referee that the visualization of the CV stained biofilms could improve the presentation of results, but unfortunately pictures are no longer available.

With regards to PP/ and PP concentrations:  are these values correct? …at concentrations of 10 µg mL-1. (PP/2) and 30 (PP) µg mL-1. Is PP/2 half of PP? Then, 15 µg mL-1 or PP = 20 µg mL-1

For vancomycin, PP corresponds to 30 and PP/2 to 15 mg mL-1. Mistakes have been corrected.

It is recommendable to Define ALT the first time it appears and check all abbreviations

Done

“Materials and methods” section:

In Table 1, “Gene-specific primers used for real-time RT-PCR,” it is recommended to include and list the GenBank access numbers for the selected genes in an extra column.

Done

“Discussion” section:

Have the authors investigated the direct antifungal activity of vancomycin against C.auris? It has been indicated that vancomycin inhibits the growth of planktonic and biofilm Candida spp (β-Lactam antibiotics and vancomycin inhibits the growth of planktonic and biofilm Candida spp.: An additional benefit of antibiotic-lock therapy? International Journal of Antimicrobial Agents,https://doi.org/10.1016/j.ijantimicag.2014.12.012).

There are contradictory results in the literature concerning the direct effect of antibiotics, including vancomycin on Candida spp., by the same authors (see Sidrim et al, 2015 https://doi.org/10.1016/j.ijantimicag.2014.12.012 ; de Aguiar Cordeiro et al., 2018 doi: 10.2217/fmb-2018-0019. Epub 2018 Jun 8 ). However, more consistent results have been reported in the case of a promoting effect (de Aguiar Cordeiro et al., 2018 doi: 10.2217/fmb-2018-0019. Epub 2018 Jun 8).

We also investigated the Van effect on planktonic C. auris growth in static conditions to support the hypothesis of a promoting effect in conditions of O2 limitations. Some increase in growth was found, but these results have not been included in the paper which is mainly focused on C. auris biofilm formation.

English language and style have been checked.